# A Fresh Look on Old Clothes: Laundry Smell Boosts Second-Hand Store Sales

**DOI:** 10.3390/brainsci12111526

**Published:** 2022-11-10

**Authors:** Jasper H. B. de Groot, Charly Walther, Rob W. Holland

**Affiliations:** Behavioural Science Institute, Radboud University, 6525 XZ Nijmegen, The Netherlands

**Keywords:** sustainability, consumer behavior, ambient odors, field study, sales, second-hand clothing, cognition

## Abstract

The clothing industry is one of the biggest polluters impacting the environment. Set in a sustainable environment, this study addresses whether certain ambient odors can influence the purchase of second-hand clothing. This study fulfilled three aims, increasing methodological, statistical, and theoretical rigor. First, replicating the finding that fresh laundry odor can boost purchasing behavior in a second-hand store—this time in a larger sample, using a fully counterbalanced design, in a pre-registered study. Second, assessing the effectiveness of another cleanliness priming control condition (citrus odor) unrelated to the products at hand, to test hypotheses from a hedonic vs. utilitarian model. Third, combining questionnaire data tapping into psychological processes with registered sales. The results (316 questionnaires, 6781 registered transactions) showed that fresh laundry odor significantly increased the amount of money spent by customers compared to the no smell condition, (replication) and compared to citrus odor (extension). Arguably, fresh laundry odor boosts the utilitarian value of the product at (second) hand by making it smell like non-used clothing, ultimately causing customers to purchase far greater amounts in this sustainable setting.

## 1. Introduction

Worldwide, a staggering 80 billion new clothing items are bought per year [1]. Fast fashion, the rapid production of clothes to keep up with consumer demands, causes the majority of clothing to be discarded, amounting to 92 million tons of waste [2]. The clothing disposal and production processes involve hazardous chemicals polluting drinking water, soil, and air—a major environmental concern [3]. Reducing the negative impact of the fashion industry on the global environment is therefore crucial. One solution from the consumer’s perspective is to sell and buy used clothing items, as this reduces waste from the production and disposal processes of new clothing. Second-hand clothing stores received a major increase in customers over the last decades [4,5]. However, the key is how consumers behave in these stores regarding the actual purchasing of second-hand items. Additionally, how can purchasing be influenced in this sustainable domain? This has remained a black box [6,7].

One technique known to influence consumer behavior is priming, a psychological phenomenon whereby exposure to one stimulus influences a response to a subsequent stimulus, without conscious mediation [8]. Priming is a robust phenomenon in psychology, typically using within-subjects designs (e.g., [9,10]). However, several between-subject studies such as walking more slowly after being primed with the elderly stereotype [11], or enhanced performance on answering knowledge questions after being primed with professors [12] triggered particular attention: not all priming studies yielded evidential value (e.g., [13]) and a series of non-replications sparked controversy (e.g., [14]). The initial lack of statistical power [15] and absent pre-registration of hypotheses and data analysis plans (the standard before 2013) should be overcome to substantiate which primes can reliably influence our behavior.

In addition to methodological improvements, more recent theoretical accounts like the situated inference model [16] help constrain the conditions in which primes would be effective. For instance, priming effects are most likely to occur when people misattribute the primed information to, for instance, the perception of a particular object (construal priming) [16]. Misattribution occurs only when people are unaware of the (effect of the) prime [16]. From this account, it would follow that smells are particularly potent primes [17,18], because olfaction has unique anatomical features that set it apart from other senses [19,20]. That is, smells bypass regions involved in conscious attention and language processing, and directly probe into areas related to emotions and associative memory [19,20]. The effectiveness of smell primes has been established in the lab, in Virtual Reality (VR), and in the field. The citrus-like smell of all-purpose cleaner, for instance, caused participants to clean more crumbs from the table [21], a finding later conceptually replicated with laundry odor causing more motivated handwashing in a VR setting [22]. Similar findings were obtained in a ‘noisier’ real-life setting: train passengers left the train wagon in a cleaner state when they sat in a citrus-scented train wagon compared to a non-scented wagon [23]. The question we ask here is whether smells can also prime and modify behavior in consumer settings, and one of the aims is to get a better theoretical understanding of why they do so.

### 1.1. Background: Smells and Consumer Behavior

The primes used to influence consumer behavior typically engage a variety of sensory stimuli such as music, lighting, and color [24,25,26], whereas smells have been underrepresented [27]. Gradually, the scientific interest in how odors affect consumer behavior has increased over the past three decades, culminating in a special issue on smells, well-being, and the built environment [28]. Ambient smells, odors that are diffused into a given atmosphere, are environmental cues that typically delight people (see [29] for a review). That is, smells can affect people through an emotional pathway. Consumers that enjoy a certain ambient odor spend more time in the store and pay more attention to the store brand, which is then also more easily recalled [30,31]. A bookstore filled with chocolate scent caused customers to spend more time in the store, talk longer to the staff, look at multiple books, and read more blurbs [32]. The smell of chocolate increased spending by about 5% compared to the no odor control condition. Yet, besides the emotional factor there are also cognitive factors determining how smells influence consumer behavior [29,33].

Some findings show that not only the pleasantness, but also the congruency of the smell in regard to the product category was crucial in affecting consumer behavior [33,34,35]. In the bookstore, products that were congruent with the chocolate smell were bought 5.92 times more than in the no smell control condition [32]. Smells congruent with a café (coffee, honey, and biscuit) induced higher customer satisfaction, greater intention to revisit the establishment, and improved perception of both product and service [36]. Moreover, congruent conditions strengthen customers’ memories of products [37]. Finally, store and product evaluation, customer mood, and time spent in the store were all elevated if rose maroc (labeled a ‘masculine’ smell) and vanillin (labeled a ‘feminine’ smell) were presented in the respective male and female product sections [35]; congruency (vs. incongruency) doubled the amount of money spent (USD 55.14 vs. USD 23.01) and number of items bought (1.71 vs. 0.91).

Hence, there are affective and cognitive roads from smell to consumer behavior. In an attempt to compare the two directly, de Groot [33] presented customers of a second-hand clothing store with no smell (regular store odor), a pleasant smell (vanilla-sandalwood), and an equally pleasant ‘fresh linen’ smell that would prime cleanliness. Refuting the affective model, customers in the fresh linen condition spent significantly more money (EUR 10.36) than in the vanilla-sandalwood (EUR 6.72) and no smell condition (EUR 5.67). Moreover, more customers in the fresh linen condition bought an item (23.77%) versus the other conditions (16%). Perceptually, fresh linen scent increased evaluations of the store, staff, and products, whereas vanilla-sandalwood did not. However, no evidence was obtained that these cognitive variables significantly mediated the effect of ambient scent on spending, which could be due to low statistical power given the small sample size (*N* = 57).

The affective–cognitive dichotomy may, however, not be a precise account of how customers are persuaded by a smell. According to Batra and Ahtola [38], consumers purchase goods for two basic reasons: (1) affective hedonic gratification (from sensory attributes); and (2) utilitarian reasons derived from the functions performed by the product [39]. Utilitarian reasons go beyond mere explanations of congruency or semantic priming. Thus, a particular smell could increase the value or use of a product. In the case of second-hand clothing, a fresh laundry smell would actually be incongruent with the typically associated musty smell; yet, this odor increases the value of a second-hand clothing item through increased perceptions of hygiene [33], making a purchase more likely. Other odors, such as citrus, semantically prime the concept of cleanliness [21,23] of the store surroundings, but expectedly do not increase the value of the clothing product per se. As goal-directed actions are related towards the products, we therefore expect a fresh laundry odor-specific increase in the value of the second-hand product, if the utilitarian model holds.

### 1.2. The Present Research

This study will increase our insight into the impact of odors on consumer behavior in a sustainable setting, which is important societally to decrease the environmental impact of the fast fashion industry, and scientifically to understand how exactly odors influence us in the real world. The triple aim is to: (1) replicate the findings of de Groot [33] regarding the impact of fresh laundry odor on purchasing behavior in a pre-registered study with a large sample and a fully counterbalanced design; (2) use a control condition (citrus odor) allowing us to test the utilitarian vs. priming hypothesis; (3) gain insight in consumer behavior through questionnaires tapping into perceptual, cognitive, and affective processes and through factually registered sales. The setting is a second-hand clothing store in the city of Nijmegen, the Netherlands.

Regarding the first goal, direct comparisons between the fresh laundry smell and the no odor condition constitute confirmatory analyses in an attempt to replicate prior research findings [33]. The first hypothesis states that a fresh laundry smell will increase the amount of money spent by customers in a second-hand clothing store (vs. no odor). The second hypothesis states that the fresh laundry smell will enhance customers’ store evaluation (vs. no odor). The third hypothesis states that store evaluation will mediate the influence of fresh laundry smell on the amount of money spent by customers in a second-hand clothing store (vs. no odor) [33]. No support for the third hypothesis was obtained in prior research [33]; yet, the current research is properly powered and has a larger sample.

Concerning the second goal, an important concern of second-hand store customers is the hygiene of the second-hand clothing [33]. However, it has remained unknown how broad or specific this concept is when it comes to odors cueing cleanliness. Aside from the previously used laundry odor [22,33], we use a citrus-like smell that is also semantically congruent with cleanliness [21]; yet, it expectedly has no utilitarian benefit when it comes to increasing the value of the second-hand clothing product. In other words, as the hygiene of the second-hand clothes is a concern, priming customers with general cleanliness of the environment through citrus odor will probably have no effect on the aforementioned outcome measures, other than potentially increasing affective hedonic gratification [38]. However, it could also be that semantic priming with the general concept of ‘cleanliness’ through citrus odor is sufficient to boost sales. The citrus condition-specific outcomes form the novel, exploratory part of this research.

As per the third goal, purchasing figures obtained via questionnaires (cf. [33]) will be supplemented with results from the cash registry, to track all transactions rather than just those reported by volunteering customers in the store. An exploratory time-based analysis will be performed on these data to track the effectiveness of smells over time.

## 2. Materials and Methods

### 2.1. Participants and Design

Our estimated participant numbers were based on a small-to-medium effect size from a comparable study testing the effect of laundry odor (vs. no smell) on money spent in a second-hand store, mediated by a positive general impression of the store [33]. Using this effect size, *f*^2^ = 0.079, α = 0.05, and 80% power, G*Power 3.1 [40] indicated a required sample size of 102 per condition. Customers that made at least one purchase were asked to voluntarily fill in a short questionnaire before exiting the store. See Figure 1, for a flow diagram showing participant demographics. Most participants (*N* = 270) performed the survey in Dutch (85.4%), the remaining 14.6% (*N* = 46) in English. Participants received a 10% discount coupon from the store they were purchasing from as an incentive.

The study had a between-subjects design, with odor being the sole experimental variable (3 levels: fresh laundry, citrus, no smell).

### 2.2. Materials

#### 2.2.1. Odors and Diffusion

Odor stimuli were created by the company Iscent (Zeewolde, The Netherlands) experienced in distributing scents in stores, hotels, and restaurants. The fresh laundry odor is made up of the following molecules: 2-ethyl-4-(2,2,3-trimethyl-3-cyclopenten-1-yl)-2-buten-1-ol, benzyl ortho hydroxy benzoate, 6,7-dihydro-1,1,2,3,3-pentamethyl-4(5h)-indanone (cashmeran), 3,7-dimethyl-6-octen-1-ol, coumarine, tricyclo decenyl acetate, 2-methyl-4-isopropyldihydrocinnam aldehyde, 2,6-dimethyl-7-octen-2-ol, hexyl cinnamic aldehyde alpha*, n-hexyl ortho hydroxy benzoate, ionon beta, iso e super, 2h-pyran-4-ol, tetrahydro-3-, lilial, limonene*, hydroxy methyl pentyl cyclohexene carbaldehyde*, alpha-iso-methylionone, nerolidol, p-methylanisole, 4-tert-butylcyclohexyl acetate*, etrahydro-4-methyl-2-(2-methylpropen-1-yl) pyran, trichloromethyl phenyl carbinyl acetate, 6-acetyl-1,1,2,4,4,7-hexamethyltetraline, 2-tertiarybutylcyclohexyl acetate (Iscent, Zeewolde, the Netherlands). Compounds marked with an asterisk were also present in the “fresh linen” scent that proved effective in prior research [33]. The citrus odor is made up of the following molecules: 2-tertiarybutylcyclohexyl acetate, alpha-methyl-1, 3-benzodioxole-5-propanal, 3,7-dimethyl-6-octen-1-ol (dl-citronellol), 2-methyl-1-phenylpropanol-2, ethyl acetate, trans-3,7-dimethyl-2,6-octadien-1-ol (geraniol), 3,7-dimethyl-2,6-octadien-1-yl-acetate, (e)-4-(2,6,6-trimethylcyclohex-1-en-1-yl)-but-3-en-2-one, 7-acetyl-(1,8)-octahydro-1,1,6,7-tetramethylnapthalene, limonene, linalool, linalyl acetate, 2,2-diemthyl-3-(3-methyl-phenyl)-propanol, neryl acetate, 2-phenylethanol, alpha-pinene, l-beta-pinene, 2-mentha-1,4(8)-diene, 4-methyl-3-decen-5-ol (Iscent, Zeewolde, The Netherlands). In the ‘no smell’ condition, customers experienced only the regular second-hand clothing store odor (i.e., no smell was distributed).

Scents were diffused via a specialized 200 × 200 × 100 mm smartphone-controllable diffusion system: ‘Scentcube’ (Iscent, Zeewolde, The Netherlands) (Figure 2), which can diffuse odors in an area of 750 m^2^ (the second-hand store being 150 m^2^). Scentcubes were placed inconspicuously above the entrance ~2 m high and next to the store’s ventilation system, to decrease the likelihood of participant awareness of the manipulation. A small fan was installed to distribute the scent equally throughout the store. Each scent (400 mL of fresh laundry odor, citrus odor) was placed into an individual Scentcube, to prevent cross-contamination. The intensity of both machines was set to the same level (3 on a scale from 1–10). The Scentcubes were set to start 1 h prior to the opening of the store to ensure the scent was distributed throughout the store before a customer entered. Customers were exposed to the scent during their stay in the store and while filling out the questionnaire in the store. During the experiment, the door of the store was left open to increase air flow, further aiding the spread of the scent inside (and outside) the store.

#### 2.2.2. Measures

A 16-item questionnaire (see Appendix A) was used to measure on 10-point Likert scales participants’ mood (valence: 1 = “very negative”–10 = “very positive”; arousal: 1 = “very low”–10 = “very high”), store evaluation (general impression: 1 = “very negative”–10 = “very positive”; cleanliness: 1 = “not at all”–10 = “very much”), and product evaluation (quality: 1 = “very negative”–10 = “very positive”; cleanliness: 1 = “not at all”–10 = “very much”); these questions were selected based on prior research [33,41], as these factors could be affected by smell and influence whether consumers purchase second-hand clothing. Next, participants were asked how much money they spent, what kind of products they bought (shirts, shoes, underwear, jackets, pants, accessories, blouses, other), how many products they bought, and how likely it is that they will visit this store again (1–10: “very unlikely”–“very likely”) [35,36]. Demographic questions on gender and age were also included. On items 13–16 (on a final, separate page of the questionnaire to prevent participants from changing their answers to the preceding questions), participants were asked whether they noticed anything unusual in the store (multiple answers possible: color or light, sound, smell, particular message, nothing unusual), they rated their overall smell ability (1–10: “very bad”–“very good”), were asked to describe the smell in the store (or choose the option that the store had not particular smell), and rate the smell on the following aspects: pleasantness, cleanliness, appropriateness, intensity, and familiarity (1–10: “very low”–“very high”). The questionnaire was available in Dutch and English, in print and online (using Qualtrics).

In addition to self-reported spending (also used in [33]), the store provided data from the cash register system, which automatically saves every monetary transaction without collecting any personal data, such as credit card information. The data included the amount of money spent, the number of items purchased, and the exact date and time of the purchase. Objective transaction data could not be directly linked to questionnaire data, but only to the smell condition and time of day. The advantage of using the cash register system is that a wealth of data could be collected with little effort, while the cost is that only assumptions can be made about the participant’s (smell) abilities and psychological processes.

### 2.3. Procedure

The experiment took place during 10 consecutive weeks (Monday–Saturday) in a second-hand clothing store in the city of Nijmegen, the Netherlands. For privacy reasons, the name of the store is not disclosed. Odor conditions (fresh laundry, citrus, no smell) changed daily based on a fully counterbalanced design. Four closing days due to religious holidays were repeated in week 10, which included two extra test days for the control condition to reach sufficient participants as per our power analysis. Store employees (who did not know the study’s hypotheses) were instructed to ask every purchasing consumer to take part in the study. As this approach far from succeeded in reaching the powered number of participants after 6 weeks, one of the researchers (C.W.) actively recruited consumers in the store until week 10. Participants were asked for their language of choice (Dutch, English) and to either fill out a printed questionnaire or scan a QR code located at the counter for the online version. At the start of the survey, participants gave informed consent; their participation was voluntary, anonymous, and stopping at any time during this 2-min survey was permitted. At the end of the survey, participants were informed about the study’s hypothesis and participants had the option for final questions or remarks.

### 2.4. Analysis

All data, materials, and the pre-registration are permanently stored on: https://osf.io/ar8td/ (accessed on 23 September 2022).

Data preparation: Online questionnaire data were retrieved from Qualtrics and imported in IBM SPSS 27; physical versions were manually inserted into the same SPSS file. After applying the exclusion criteria (see Participants and design), the questionnaire data included 316 participants (fresh laundry odor: *N* = 104; citrus odor: *N* = 100; no smell: *N* = 112). As a Shapiro–Wilk test indicated that data were not normally distributed, outliers were identified using a very conservative threshold of 3 units of Median Absolute Deviation, the most robust procedure in the presence of outliers [42]. Outliers were then winsorized, which entails the reassignment of values of 1 unit above (or below) the value nearest to the outlier that according to MAD analysis is not an outlier [43]. Winsorizing increases power by maintaining all data points while still reducing the outlier’s influence [44]. We would like to note that while standard (e.g., [22,45,46]), we had not explicitly included outlier analysis in the current pre-registration. We therefore report the confirmatory analysis with and without outliers.

The second-hand store also shared anonymous monetary transactions in a Microsoft Excel file. Of these initial 7109 transactions, 6781 (95.4%) could be used (the list also contained negative numbers, meaning that customers returned items): fresh laundry odor, *N* = 2202; citrus odor: *N* = 2035; no smell: *N* = 2544. To chart the effects of smell over time (per hour) in an exploratory fashion, the variable time point was added. Again, MAD outlier analysis was applied, before the datafile was imported into SPSS.

Statistical analysis: The first and second hypotheses were tested with a one-way ANOVA, with odor as the sole between-subjects factor. Specific contrasts compared the three odors. Comparisons between fresh laundry smell and the no odor condition is confirmatory [33], whereas any comparison with the citrus smell is exploratory. Lastly, a mediation analysis was performed testing the effect of odor on self-reported amount of money spent mediated by store evaluation (combined items general store impression and store cleanliness) [33], using the PROCESS macro for SPSS (Version 3.5.3) following the method of Hayes [47]. Within the PROCESS macro, model 4 was selected to test hypothesis 3 with simple mediation. Unstandardized indirect effects were computed for each of 5000 bootstrapped samples, and the 95% confidence interval (CI) was computed. Mediation was present if the 95% CI did not overlap with 0.

## 3. Results

### 3.1. Hypothesis Tests

The first hypothesis was that exposure to a fresh laundry odor would cause people to spend more money in a second-hand clothing store compared to the no smell condition (regular store odor) and a citrus control odor. A one-way ANOVA was performed, with condition (3 levels: fresh laundry, no smell, citrus odor) as between-subjects factor, and the self-reported amount of money and monetary transactions as dependent variables. Regarding self-reported money spent, the effect of odor was significant, *F*(2,313) = 3.81, *p* = 0.023, η^2^ = 0.02. Planned contrasts indicated that, as expected, customers spent significantly more money in total in the fresh laundry condition (*M* = EUR 17.52, *SD* = EUR 13.70) compared to the no smell condition (*M* = EUR 13.60, *SD* = EUR 9.69), *p* = 0.011 (with outliers: *p* = 0.035), *d* = 0.33, 95% CI [0.90, 6.94], but also compared to the citrus odor condition (*M* = EUR 14.06, *SD* = EUR 10.08), *p* = 0.029 (with outliers: *p* = 0.124), *d* = 0.29, 95% CI [0.35, 6.57] [laundry vs. both controls: *t*(313) = 2.73, *p* = 0.007 (with outliers: *p* = 0.036)] (Figure 3A). No significant differences were present between the no smell and citrus odor condition, *t* < 1. The data of the registered transactions showed a similar pattern, but not a significant effect of odor, *F*(2,6860) = 2.34, *p* = 0.096; yet, there was a significant difference between the factual amount of money spent in the laundry odor condition (*M* = EUR 7.95, *SD* = EUR 6.07) vs. both the citrus odor (*M* = EUR 7.61, *SD* = EUR 5.91) and no smell condition (*M* = EUR 7.62, *SD* = EUR 5.90), as indicated by a planned contrast: *t*(6860) = 2.16, *p* = 0.031, *d* = 0.06. Including outliers into this analysis caused only the difference between fresh laundry odor and no smell to remain intact, *p* = 0.049. Next, an exploratory analysis was performed on the amount of money spent over time per odor condition (Figure 3B). The only significant differences between fresh laundry odor and the control conditions (no smell, citrus odor) emerged in the 4th and 5th hour of odor exposure, from 1–2 p.m., *t*(6757) = 2.19, *p* = 0.029, and from 2–3 p.m., *t*(6757) = 2.38, *p* = 0.017.

To test hypothesis 2, that fresh laundry odor increases the store evaluation (vs. other conditions), a one-way ANOVA was conducted on the questionnaire items ‘general store impression’ and ‘store cleanliness’. There was neither a significant effect of odor on general store impression *F*(2,313) = 0.184, *p* = 0.832, nor on store cleanliness *F*(2,313) = 0.374, *p* = 0.689. Means were on the high end of the scale for all conditions regarding general store impression (laundry odor: *M* = 8.29, *SD* = 1.13; citrus: *M* = 8.20, *SD* = 1.04; no smell: *M* = 8.27, *SD* = 1.10) and store cleanliness (laundry odor: *M* = 8.19, *SD* = 1.13; citrus: *M* = 8.24, *SD* = 0.92; no smell: *M* = 8.31, *SD* = 1.02)

To test the third hypothesis, that store evaluation mediates the effect of fresh laundry smell on the amount of money spent per consumer (vs. no smell), a simple mediation analysis was used. Neither the pathway from odor condition to store evaluation (*b* = −0.05, *t* = −0.40, *p* = 0.687 [95% CI: −0.29, 0.19]), nor the pathway from store evaluation to money spent was significant (*b* = 1.62, *t* = 1.84, *p* = 0.067 [95% CI: −0.12, 3.36]). The direct path from condition to money spent (*b* = 4.00, *t* = 2.51, *p* = 0.013 [95% CI: 0.85, 7.15]) was significant, while the indirect effect of condition on money spent via the mediator store evaluation was not significant (*b* = −0.08 [95% CI: −0.58, 0.44]).

### 3.2. Additional Analyses

Despite significant odor-based differences in money spent, the number of items bought did not differ: *F*(2,7101) = 1.91, *p* = 0.149 (fresh laundry odor: *M* = 1.14, *SD* = 0.46; citrus: *M* = 1.17, *SD* = 0.54; no smell: *M* = 1.16, *SD* = 0.59). Moreover, there were no significant differences in judgments of product quality (*F* < 1), product cleanliness (*F* < 1), subjective feeling *F* < 1, subjective energy level, *F*(2,313) = 1.41, *p* = 0.244, and the intention to revisit the store: *F* < 1.

As the abovementioned results could have been driven by smell awareness, the one-way ANOVA on money spent was performed for the 269 individuals who reported being unaware of a particular smell in the store. The significant effect of odor remained intact, *F*(2,266) = 4.48, *p* = 0.012, η^2^ = 0.03 (fresh laundry, *n* = 79: *M* = EUR 18.34, *SD* = EUR 14.16; citrus, *n* = 91: *M* = EUR 13.96, *SD* = EUR 10.17; no smell, *n* = 99: *M* = EUR 13.62, *SD* = EUR 9.86). Analyzing the 39 customers that did report smell awareness yielded no significant differences between conditions, *F* < 1 (fresh laundry, *n* = 24: *M* = EUR 15.37, *SD* = EUR 12.10; citrus, *n* = 9: *M* = EUR 15.08, *SD* = EUR 9.55; no smell, *n* = 6: *M* = EUR 15.00, *SD* = EUR 10.71).

Customers also reported on various properties of the smell, namely intensity, pleasantness, cleanliness, appropriateness, and familiarity. Between odor conditions, there was a significant difference in self-reported odor intensity, *F*(2,306) = 5.19, *p* = 0.006, η^2^ = 0.03. Customers perceived the fresh laundry odor as more intense (*M* = 6.25, *SD* = 1.74) than the citrus odor (*M* = 5.40, *SD* = 2.09) and the regular store odor in the no smell condition (*M* = 5.65, *SD* = 1.88). The variables pleasantness (*F* < 1), cleanliness (*F* < 1), appropriateness (*F* < 1), and familiarity (*F* < 1) were not significantly different between odor conditions. Controlling for odor intensity in an ANCOVA on money spent with factor odor condition, the effect of the covariate odor intensity was significant, *F*(1,305) = 5.42, *p* = 0.020, η^2^ = 0.02; yet, the intensity covariate did not eradicate the significant difference between fresh laundry odor and the control conditions on money spent, *t*(305) = 2.02, *p* = 0.045.

The sample consisted of mainly females (78%). Analyzing only the male subset (*N* = 70) left the significant effect of odor on self-reported money spent intact, *F*(2,67) = 5.13, *p* = 0.008, η^2^ = 0.13 (fresh laundry, *n* = 16: *M* = EUR 25.31, *SD* = EUR 15.37; citrus, *n* = 25: *M* = EUR 16.34, *SD* = EUR 10.08; no smell, *n* = 29: *M* = EUR 14.39, *SD* = EUR 9.34). If anything, the effect was stronger in males, changing the effect size from small-to-medium to large.

## 4. Discussion

The overarching goal of this study was to investigate the influence of odors on consumer behavior in a sustainable second-hand store setting.

### 4.1. Hypothesis Tests

The first hypothesis stated that a fresh laundry smell increases the amount of money spent by customers in a second-hand clothing store (confirmatory: vs. no odor; exploratory: vs. citrus odor). The results support this hypothesis, replicating and extending previous research [33]. In that study [33], the self-reported money spent by participants was significantly higher when customers were exposed to a ‘fresh linen’ scent compared to an equally pleasant control odor (vanilla-sandalwood), and compared to the regular store odor (no smell condition). Regarding the present research, the self-reported amount of money spent (*N* = 316) and actual transactions (*N* = 6781) were recorded. This is a large sample to demonstrate purchasing behavior compared to prior studies reporting odor-specific effects in around 200 customers [32,35]. Still, despite the larger sample of objective transactions, the effects were weaker than those obtained for the self-reported money spent, which seems counterintuitive. A possible explanation for the difference in results could be that participants indicating a poor sense of smell were excluded from the questionnaire sample (~4% of the initial sample), as per our pre-registration (the non-smeller sample was too small to perform a reliable analysis to substantiate this point); yet, it was not possible to apply these criteria to the anonymous transactions (a separate dataset), which could have included a similar portion of poor smellers for a variety of reasons (e.g., anosmia/hyposmia, blocked nose due to a cold, COVID-19, wearing a facemask) reducing the sensitivity of that sample to the odor manipulation. Future research having a similar set up may however perform power analysis using the (lower) effect size from the objective transaction data and thereby automatically consider that a subset of the sample is a non-smeller. Another reason for the smaller transaction-effect may be that these data included customers that spent little time in the store, which reduces the impact of the odor on their behavior. Reversely, participants filling out the questionnaires may have been more patient, spending an increasing amount of time in the store, and were thus more likely to be influenced by the smell [48]. According to Emsenhuber [49], more complex associations such as scents linked to a product property and its values can take up to several minutes to process. As the time spent in the store was not measured in the current study, this is only an assumption. In sum, only a fresh laundry odor proved effective in evoking more self-reported and objective purchasing behavior in the second-hand clothing store.

The second hypothesis stated that the fresh laundry smell enhances consumers’ store evaluation (confirmatory: vs. no odor; exploratory: vs. citrus odor). Here, the results did not support the hypothesis, which is at odds with previous research [33,35,50]. Compared to the control conditions, the fresh laundry smell neither increased the general store evaluation, nor the evaluation of store cleanliness. When comparing the present general store evaluation findings (*M* = 8.22, *SD* =1.15) to the most comparable research (*M* = 6.10, *SD* = 0.66) [33], the generally higher values obtained here might have created a ceiling effect, limiting the ability to find a boosting effect of scent on store evaluation in the present setting.

The third hypothesis stated that store evaluation mediates the influence of fresh laundry smell (vs. no odor) on the amount of money spent by customers. Despite having sufficient statistical power, this hypothesis was not supported. This finding is, however, consistent with the null findings obtained in prior research; yet, there, a smaller sample was used, which initially limited statistical power to detect a mediation effect [33].

### 4.2. Exploratory Findings

The present study provided the first insights into the temporal dynamics of consumer spending as a function of odor. Notably, between 1 and 3 p.m., sales were significantly higher in the fresh laundry condition compared to the citrus odor and no smell condition. A possible explanation is that around this time, more customers are present that do unplanned purchases—they are hedonistic shoppers with fewer time constraints [51]. It is apparently under these conditions that smells are most effective in influencing consumer behavior. Scent primes may be less effective when consumers have a clear goal and intention of what they want to purchase [52]. These combined reasons may explain why consumers were persuaded more by the laundry smell in the early afternoon. It also notes that smells do not do a ‘magic trick’ when it comes to influencing customers. Not all customers are influenced at every time point. Instead, the effectiveness of the smell depends on a complicated interplay between environmental variables and personal characteristics, states, and goals.

Most participants (85%) reported no awareness of a smell that could have influenced their behavior in the store. It has been noted that consumers should be unaware of the scent to increase the effectiveness of the odor on spending behavior [16,34,53]. Excluding the aware participants did not alter the conclusions—fresh laundry odor still significantly increased sales. If anything, the effect size increased. Although most people were unaware of the smells, participants did rate the intensity of the fresh laundry scent as significantly higher compared to the other conditions. As both fresh laundry odor and citrus odor were diffused at the same level of intensity from the Scentcubes, and Iscent manufactured the scents with the same subjective intensity, this seems to be a solely subjective impression by the participants. One possibility is that participants associated fresh laundry odor more with the second-hand clothing and thus, unconsciously, rated the intensity higher compared to the citrus smell [54]. Importantly, any differences between the scent conditions were subjective and not objective, and therefore cannot pose a confound.

### 4.3. Limitations, Recommendations, and Implications

The current study has a number of limitations. One is a potential constraint on generality posed by the characteristics of the sample [55]. The majority of the sample (78%) were female, like in prior comparable research (93% [33]), and had high intentions to revisit the store, even in the control condition. Research has shown that women generally visit more stores [56] and second-hand clothing stores in particular [57]. A meta-analysis revealed that females have a better smell than males, but only slightly so [58], from which it could be argued that females are more susceptible to the influence of odors. However, (1) smells typically have stronger effects on us when we are not aware of them [17,18]; and (2) males are suspected to have the same associations with second-hand clothing items (unhygienic) that could prevent them from a purchase, which could be overridden by a fresh laundry smell. In prior research [33], it was considered unlikely that males would remain unaffected by a ‘fresh linen’ scent; yet, due to the small sample size, this assumption could not be tested. Analyzing the male subsample in the present study (*n* = 70) revealed a strong effect of fresh laundry odor on (self-reported) money spent. Hence, generality when it comes to gender is not a valid concern. We do recommend testing the generalizability of these findings to non-Western samples, and to include mixed audiences in stores that offer both fast fashion and sustainable items to see if also those individuals not typically inclined to buy second-hand clothing can be persuaded by a smell.

Another limitation, which could not be verified with quantitative data, is the associations customers had with second-hand clothing and the fresh laundry and citrus smell. Customers did provide qualitative labels for the smells they experienced. In the fresh laundry condition, 28% reserved a positive label (e.g., fresh, pleasant, perfume, clean; vs. citrus condition: 15%; no smell condition: 17%), 16% reserved labels like “musty”, “old clothes”, “used” or “second-hand”; citrus: 12%; no smell: 12%), and 6% found the smell to be “neutral” (vs. citrus: 17%; vs. no smell: 9%). A large portion, 26%, indicated experiencing no smell at all (vs. citrus condition: 40%; no smell condition: 49%). In the present research, no pre-test was performed to investigate whether the citrus odor was indeed associated with cleanliness; its selection was based on prior research where the scent made cleaning-related words more accessible [21] and induced cleaning behavior [21,23]. The fresh laundry smell has a certain overlap with the ‘fresh linen’ scent used in prior research [33], where it was pre-tested for its association with cleanliness and hygiene. On a molecular level, four compounds were identical between the fresh laundry odor shown effective before [33] and the one used in the current study: hexyl cinnamic aldehyde alpha, limonene, hydroxy methyl pentyl cyclohexene carbaldehyde, and 4-tert-butylcyclohexyl acetate. It seems tempting to further investigate these four molecules to see how critical they are in creating a sensation of a ‘fresh laundry’ scent; however, because specific anosmia (i.e., the inability to smell a specific odorant) is so widespread, the blend has the advantage of containing all potentially odor-active molecules for anyone able to perceive the complex smell of fresh laundry [59].

The third limitation concerns the lack of experimental control, for instance over the dispersion of the smell in the store. We have no objective indicators that smell molecules were dispersed equally well in the store, and that levels of molecules were constant over time. The more low-end solution is to collect pilot data on subjective odor pleasantness and intensity, whereas a more high-end option is to use a photo-ionization detector to objectively quantify over time the number of molecules that are dispersed in different locations in the store, to ensure that the odor stimuli have similar concentrations. Other ideas to get a better grip on the effectiveness of smell on consumers could be to track the (approximate) time participants spend in the store (via self-report) and to register the number of people who enter the store and buy products (conversion rate). In this environment with relatively little control, yet in an experiment with sufficient power, we obtained small effect sizes for the fresh laundry odor. Hence, the fresh laundry odor may help individuals to buy more (high quality) second-hand clothing, but future research should assess whether this also reduces the number of new clothing items bought to substantiate environmental impact.

Aside from having its limitations and listing recommendations, the current study bears theoretical implications. Past research on smell-based consumer behavior can be roughly divided into taking a more affective/hedonic or cognitive viewpoint. From the hedonic view, it is argued that certain smells affect consumers because they are pleasant [31,32]. From the cognitive view, it is argued that smells can influence our behavior by activating particular semantic knowledge [34,35,36,37]. In some cases, affective and cognitive variables are directly compared [33,41]. For instance, de Groot [33] found that a pleasant smell (vanilla-sandalwood) was insufficient to increase the amount of money spent in a second-hand store (vs. no odor); only a fresh laundry-like smell associated with cleanliness and hygiene could boost sales. This seemingly supports a cognitive account of semantic priming (e.g., [18,21,22]. However, not every odor associated with cleanliness can influence consumer behavior in a second-hand store setting, as the present study has shown. By comparing a citrus scent (associated with cleanliness but not with clothes) to a fresh laundry scent (associated with cleanliness and clothes), we found that only the latter odor was effective. Arguably, the fresh laundry smell increased the subjective value of second-hand clothing items by overriding initial perceptions of the clothing being unhygienic (cf. [33]). Consistent with a model of value-based decision making [60], customers initial valuation of a second-hand clothing item could perhaps be updated by a specific smell that enhances the value of the second-hand clothing item by making it appear a new, unused item. Hence, for a smell to be effective in a (sustainable) consumer setting, it needs to be categorically tied to the product that is being bought, arguably giving it the same properties and functionality as new clothing items. Indeed, aside from hedonic reasons consumers have instrumental or utilitarian reasons for buying products [38,39]. A good fit between scent and product seems crucial.

## 5. Conclusions

Previous research on smells affecting consumer behavior have targeted stores (e.g., bookstore, wine store, or retail) for which sustainability was not a core theme [29]. The current pre-registered study provided relevant insights into how odors influence (subjective and objective) consumer behavior in a sustainable setting. Although cleanliness is associated with both a fresh laundry odor [22,33] and a citrus-like odor [21,23], we report evidence—using a well-powered experiment with fully counterbalanced design—that customers only spent more money on second-hand clothing when exposed to the fresh laundry odor. These findings are a replication and extension of earlier work [33] and seemingly fit a utilitarian (vs. hedonic) view on consumer behavior. The fashion industry is one of the biggest polluting industries [61], and a cheap and effective solution to facilitate sustainable behavior can be mere molecules that give a fresh ‘look’ on old clothing.

## Figures and Tables

**Figure 1 brainsci-12-01526-f001:**
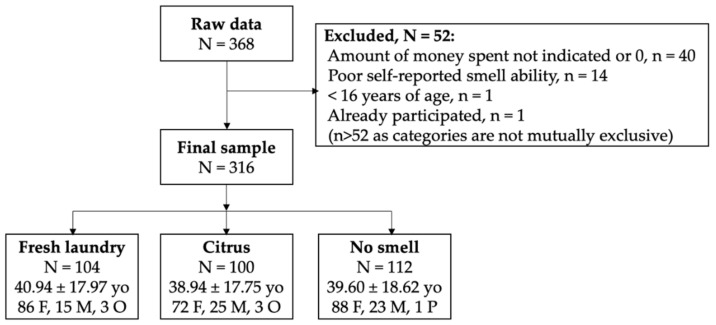
Flow diagram showing included and excluded participants and demographics. N = number of participants; yo = age in years; F = female; M = male; O = other; P = prefer not to say.

**Figure 2 brainsci-12-01526-f002:**
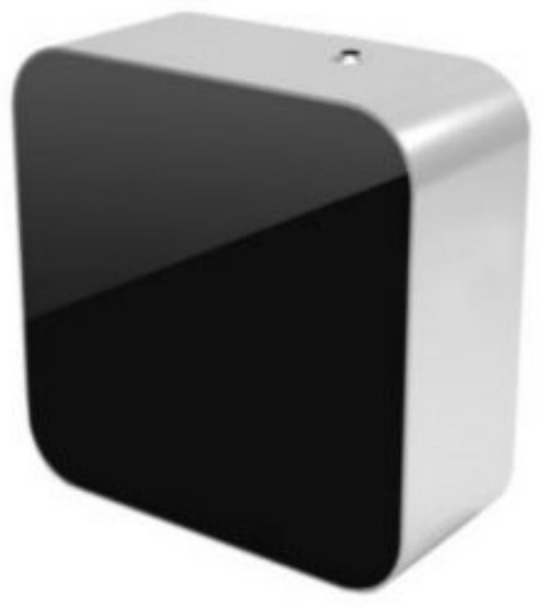
Odor diffusion system (Scentcube, Iscent, Zeewolde, The Netherlands).

**Figure 3 brainsci-12-01526-f003:**
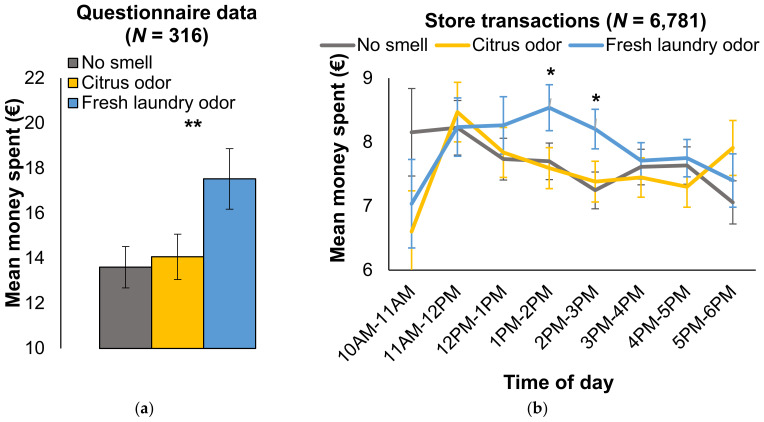
Effect of odor (no smell, citrus odor, fresh laundry odor) on money spent in second-hand store, based on (**a**) questionnaire data and (**b**) the store registry, plotted over time. Error bars ± 1 S.E. * *p* < 0.05, ** *p* < 0.01.

## Data Availability

All data, materials, and the pre-registration are permanently stored on: https://osf.io/ar8td/ (accessed on 23 September 2022).

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
