# Peer review of "A Fresh Look on Old Clothes: Laundry Smell Boosts Second-Hand Store Sales"

_brainsci, 2022, doi:10.3390/brainsci12111526_

Round 1

Reviewer 1 Report

The paper is interesting. It deals with a topic that interests the reader and highlights how the consumer can be influenced by factors related to sensoriality.

Small changes are suggested.

In the abstract there is no real objective, or in any case the authors should underline how the study can be a contribution to the already existing literature on this subject.

In the methods, at lines 166-171, it is recommended to give a different structure when listing the sample. It would be better to use a table or something graph that best presents the data in general.

At the level of paper design, I suggest using paragraphs and sub-paragraphs in abundance that untie the discussion and make reading and fluency difficult. Especially in the methods, they seem too many.

Where possible, update and supplement the bibliography

Reviewer 2 Report

I find the work by Groot et al entitled “A fresh look on old clothes: Laundry smell boosts second-hand 2 store sales” interesting. Overall it is well written and easy to follow, even for a reader who does not deal with these topics. I have only a few considerations to make.

11)      L36 and L40: I find this way of writing the refernces rather strange ... why do the authors refer to other works? why don't they put all the references together? why don't they only insert those that they think are really useful?

22)      L63-65: The authors declare that they want to try to identify a possible mechanism for the action of odors on consumer behavior, however I have found nothing of the kind in the discussion.

33)      L109: “…functions (also see [39]).” compared to what?

44)      L380-388: I find it pointless and boring to repeat the aims of the study so extensively at the beginning of the discussion

55)      L402-409: I find this may have been a serious problem. How could this be solved?

66)      L409-411: It is not clear to me whether in the transactions considered there are also those of those who have not completed the questionnaire.

77)      L499-500: I only partially agree with this suggestion: the authors do not consider specific anosmia. The blend has the advantage of containing all potentially odor-active molecules for anyone able to perceive the complex smell of fresh laundry. See this work: https://doi.org/10.1016/j.physbeh.2020.112820

88)      L545-549: this part is redundant.
